# The Historical Aspect of the Impact of Zn and Pb Ore Mining and Land Use on Ecohydrological Changes in the Area of the Biała Przemsza Valley (Southern Poland)

Kazimierz Różkowski [1] , Jacek Różkowski [2] and Oimahmad Rahmonov [2,*]

1    Faculty of Civil Engineering and Resource Management, AGH University of Krakow, al. Adama Mickiewicza 30, 30-059 Kraków, Poland; kazik@agh.edu.pl

2    Institute of Earth Sciences, Faculty of Natural Sciences, University of Silesia in Katowice, Będzińska 60, 41-200 Sosnowiec, Poland; jacek.rozkowski@us.edu.pl

\*    Correspondence: oimahmad.rahmonov@us.edu.pl

**Abstract:** The article presents the impact of Zn and Pb ore mining and land use on ecohydrological changes in the area of the Biała Przemsza valley in the historical aspect, with particular emphasis on the period 1941–2021. GIS was used to analyse the maps to identify spatial and temporal changes in land use. The following trends could be observed in the spatial development of the Biała Przemsza valley: rapid urbanisation and industrialisation in the 21st century, marked reduction in the surface area of arable land and sands, and dynamics of the increase in the surface area occupied by forests. Notable changes occurred in the plant ecosystems between 1941 and 2021 due to land management. Groundwater level lowering due to mining activity resulted in the change from wet meadows to fresh or dry xerothermic grasslands, while forest cover increased by 4% within the catchment area. The hydrographic network evolved. After the commissioning of the Zn-Pb ore mines Olkusz and Pomorzany in the 1960s and 1970s, a regional depression cone with an area of 400 km$^2$ was formed, while the maximum groundwater inflows to the mines reached 360 m$^3$/min. Changes in the hydrodynamic conditions have resulted in changes in the hydrogeochemical regime of the Triassic aquifer manifested by increased levels of, e.g., SO$_4$ and Cl. Wastewater with lignosulphonate compounds from the paper factory caused periodic degradation of some of the water flowing into the Pomorzany mine. As a result of mining activity, the character of some sections of the Biała Przemsza river changed to an infiltrating one, the quantitative depletion of groundwater resources within the range of the mine drainage cone occurred, river springs disappeared, and the flow in the river decreased. At the same time, mine water was discharged to the tributaries of the Biała Przemsza. A radical reduction in the flow of the Biała Przemsza and its tributaries occurred after the decommissioning of the Zn-Pb ore mines at the turn of 2021 and 2022.

**Keywords:** Zn-Pb ore mining; fissure and karst water; vegetation changes; Biała Przemsza River; ZGH Bolesław





## 1. Introduction

From the 12th century onwards, the Silesian-Krakow region in southern Poland was an area of exploitation of silver-rich lead ores [1–3]. In the 19th century, the increasing demand for zinc led to a significant redevelopment of Zn-Pb ore mining. In the 19th and early 20th centuries, hundreds of small open pits and underground mines exploiting Zn-Pb ores and metal-enriched tailings (Zn, Pb, Cd, Tl, As, an Sb) with areas ranging from one to several hectares were built in an area of approximately 1200 km$^2$ between Olkusz, Bytom, and Chrzanów. After World War II, three mines were commissioned, the last of which completed dewatering in 2021. The development of lead and silver ore and then zinc mining in the Olkusz region became the drivers of changes in hydrological conditions [4–7]. As early as in the 16th century, five adits for draining ore deposits were driven in the

region [8] with a total length of workings of approx. 30 km; in the 1980s, around the Zn-Pb mines, the total regional depression cone already boasted an area of 400 km$^2$ [9] (or even 500 km$^2$ according to others [10]). The maximum total inflows to the mines had already reached 360 m$^3$/min in the 1990s, and mine water was discharged mainly to watercourses in the catchment area of the Biała Przemsza [9,11]. Research on the impact of mining operations on the water environment was also conducted [5,12–18]. Gradual lowering of the drainage base to 60, 90, and finally 140 b.g.l. influenced the widening area, affecting wetlands particularly strongly within river valleys. Large depressions of the water table can reach neighboring hydrogeological units (possibly neighboring water horizons), which under normal conditions function as separate subsystems [19]. Mining and smelting of Zn-Pb-Ag ores in the 19th and 20th centuries had a major impact on Europe's natural environment. A significant environmental concern at the sites of mining and processing of zinc, lead, copper, tin, and other metal ores is the potential contamination of soil and water with toxic metals and their migration into the biotic environment [20–22]. Mining and post-mining landscapes represent an ecological niche [23–25] where a particular succession of vegetation develops in soils enriched in Zn, Pb, Cd, As, and Fe [2,26]. Botanical and pedological studies in areas affected by Zn-Pb-Fe ore mining have been conducted in Belgium [27], Germany [28], Spain [29], Slovenia [30,31], China, [32,33], Morocco [22,34], and Poland [26,35,36].

In parallel, studies were conducted on the revitalisation and remediation of metalliferous mine wastes and other anthropogenically degraded areas [37–41] in which the authors demonstrated the role of natural vegetation succession and the most appropriate methods for the remediation, revitalisation, and management of brownfield sites. Soil chemistry and mineral composition determine the development of certain plants; e.g., metallophilous ecotypes and xerothermic grasslands [28,42,43].

The period of geochemical alteration in the Olkusz region has been estimated to last some 90–100 years [44]. However, former mining wastes may be active in different geological situations, including a 200-year duration as in Belgium and the Netherlands [45] and even several thousand years as in Roman-era wastes in England [46]. The rate of development of these ecosystems depends on the degree of geochemical transformation of minerals and potentially toxic metals in the environment. The aim of this study was to present the impact of Zn and Pb ore mining and land use on ecohydrological changes in the Biała Przemsza valley region with a focus on the 1941–2021 period.

## 2. Materials and Methods

The analysis of the environmental and landscape changes in the Biała Przemsza valley area was based on the published and unpublished materials, including maps from 1941–2021 and field research. To assess the environmental changes in the studied area, the available cartographic material [47–49] and surveys from the surrounding locations were used.

Map analysis and interpretation were carried out using geographical information system (GIS) methodologies, and an interpretative sketch was created in the MapInfo and QGIS geospatial programs. Rectification was carried out as a second step, and the image was adjusted to the reference layer using control points. Topographic maps from 1993–1996 and an orthophoto map are available in georeferenced form. The results obtained from the orthophoto maps (from 2021) were verified and supplemented with additional information from the field. All topographic maps are available in the Open Regional Spatial Information System (ORSIP)—Geoportal of the Silesian Province [50]. The impact of Zn and Pb ore mining on the ecohydrological environment was presented on the basis of the published literature and unpublished archival materials [51] obtained from the Zakłady Górniczo-Hutnicze Bolesław (ZGH Bolesław Mining and Metallurgical Plant).

### 2.1. Climate and Hydrography

In the study area, the average annual temperature is 8.1 °C, in January −2.4 °C, and in July 17.8 °C [52]. In the multi-year period of 1987–2021, the total values of annual precipitation ranged from 525 mm (1989) to 1012 mm (2010), and the average precipitation for the multi-year period of 1987–2021 was 735 mm. The catchment area of Biała Przemsza is located in the province of the Polish Uplands, a macro region of the Silesian-Krakow Upland [53]. Biała Przemsza is 63.9 km in length, and its catchment area occupies 876.6 km$^2$ (Figure 1).

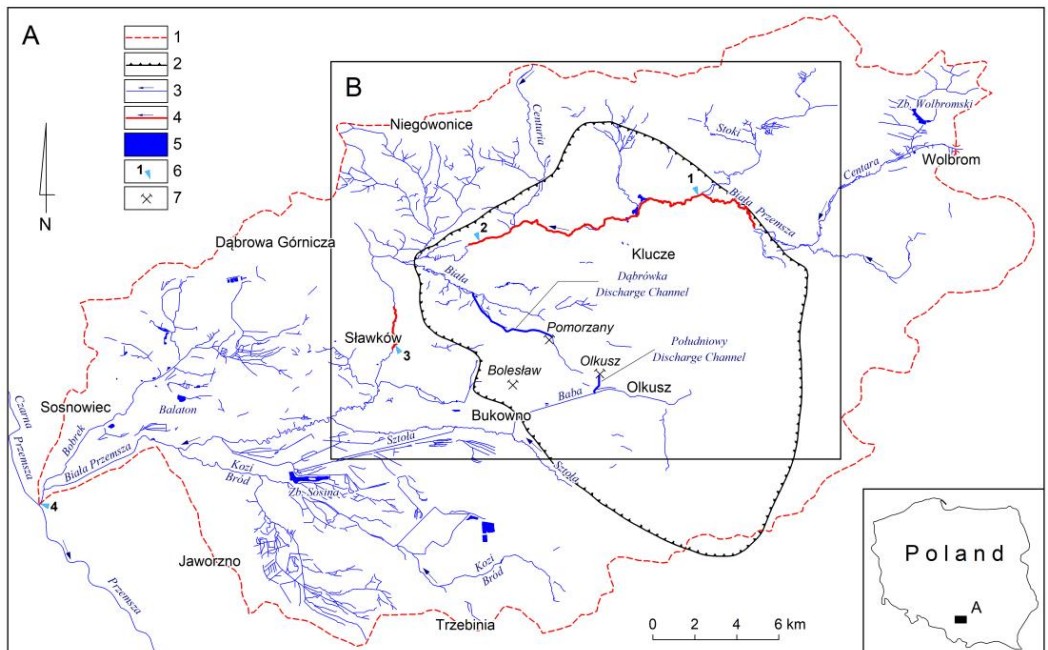

**Figure 1.** The location of the Biała Przemsza River catchment (**A**) and research area (**B**): 1—the catchment area border of the Biała Przemsza River; 2—the range of the regional depression cone; 3—rivers; 4—section with infiltration of Biała Przemsza River; 5—surface water reservoirs; 6—the water gauges; 7—mining of Zn and Pb ores.

The spring area, which is located in the Olkusz Upland, is at an altitude of 376 m a.s.l. in a peat bog near Wolbrom. The primary springs of the Przemsza River near Suska Górka have been deprived of water for years. From its springs to Okradzionów, the valley is latitudinal. On the Golczowice–Błędów section, the river flows through the Błędowska Desert. The valley, which is 200 m in width, is approx. 10 m deep into Quaternary sands underlain by Triassic formations. In this area, the Biała Przemsza strongly infiltrates deep into the ground. The effect of this phenomenon is the almost complete disappearance of the river at the western edge of the desert. Biała Przemsza is joined by its tributaries—Błędowski Stream and the left-bank Biała River (Figure 1). From Błędów, the valley continues within the Triassic formations, and in the vicinity of Okradzionów, the direction of the valley changes to the south (S), where Biała Przemsza flows in a narrow valley with steep rocky slopes through the Middle Triassic Threshold. To the south of Sławków, the river enters the Biskupi Bór Basin towards the southeast direction (SE). The valley widens and is joined by its left-bank tributaries Sztoła and Kozi Bród as well as the right-bank Bobrek. Sztoła carries water from the decommissioned Czartoryska adit. This water is captured in Maczki for water supply purposes. To the south of Olkusz, there are the Baba and the Witeradowski Stream (Figure 1), which disappear into the Quaternary sands after a few kilometres from their springs. The last 4 km of the river flows in a concrete bed. In Mysłowice at an altitude of approx. 250 m a.s.l., the Biała Przemsza flows into the Przemsza River, a left-bank tributary of the Vistula.

The Biała Przemsza River in its upper reaches to Golczowice was characterised by high flows in the winter half-year (snowmelt), while further in the Błędowska Desert section, the river infiltrated into the sandy substrate. According to Punzet [54], the average general water balance in Biała Przemsza in the years 1947–1955 from the springs to Maczki for the catchment area of 626 km$^2$ was as follows: precipitation 675 mm, surface runoff 243 mm, and balance losses 432 mm. The average flow (SQ) in Biała Przemsza was 7.3 m$^3$/s at the mouth. The average flows in the years 1964–1973 at the water gauges were: Golczowice 1.6, Błędów 2.9, and Sławków 4.2 (m$^3$/s) [4].

## 2.2. Geological Structure

Paleozoic, Triassic, Jurassic, and Quaternary deposits have been found in the geological profile of the Olkusz deposit area. The Triassic formations are represented by Bunter Sandstone formations (sandy and clay formations and dolomites), Muschelkalk formations (limestones and dolomites, conglomerates), Keuperian formations (shales, mottled clays), and Rhaetian formations (mudstones). The Middle Jurassic is represented by clayey-conglomerates and marls, and the Upper Jurassic by limestones and minor marls. The Quaternary formations are sandy and clayey with a maximum thickness of 80 m in the Przemsza fossil valley. In the tectonics, folded formations of the Palaeozoic structural stage and the Permo-Mesozoic structural stage of block tectonics can be distinguished (the horsts of Sławków-Olkusz and Klucze-Niegowonice as well as the Pomorzany Trench-Okradzionów) [4].

## 2.3. Characteristics of Zn-Pb Ore Deposits

Zn and Pb mineralisation in the Olkusz region can be found in the geological formations from the Devonian to the Jurassic. The mineralisation in the ore-bearing dolomites of the Middle Triassic (90% of the total resources) has a deposit value. The thickness of the deposit varies from 2.5 m to 6.8 m. Mineralisation occurs in the form of filling voids in the dolomite breccia, crusts, and impregnation. Generally, the mineral composition of ores is dominated by the simple paragenesis of Zn-Pb-Fe sulphide minerals [1,3,44]. In the ore minerals, apart from Zn, Pb, and Fe, there are accompanying elements: Ag, Cd, Tl, and As. The aforementioned features and the low crystallisation temperature of ores enable their classification as epigenetic, low-temperature Mississippi Valley Type (MVT) hydrothermal deposits [44]. In the mines of Bolesław, Olkusz, and Pomorzany, sulphide ores were mined along with calamine ores, which were also mined in the Bolesław mine oxidation zone. In mined ores, the content of Zn was 4.7–5.11%, while the content of Pb was 2.5–2.9% [4].

## 2.4. Hydrogeological Conditions

The documented part of the Biała Przemsza catchment area lies in the SE part of the fissure-karst-pore Major Groundwater Basin (MGB) of Olkusz-Zawiercie composed of carbonate formations of the Triassic carbonate series that are semi-confined in nature. The area of the reservoir is 1033 km$^2$, the thickness of the water-bearing formations is 20–140 m [55,56], and the specific capacity ranges from <1.0 to 550 m$^3$/h/1 mS. The recharge of the Triassic aquifer complex is direct in outcrops (parts S and W of the reservoir) and indirect through hydraulic contacts with the Quaternary, Jurassic and Paleozoic formations, and in the conditions of drainage of the aquifer complex with water infiltrating from surface watercourses. The regional recharge area of the reservoir is the hills of the Kraków Upland (400–500 m a.s.l.), while the regional basis for drainage is the river valleys of the Silesian Upland (250–300 m a.s.l.). The local directions of water flow are mainly affected by mining workings of Zn-Pb ore mines, large groundwater intakes, and geological discontinuities. The described area is located in a subreservoir associated with the Olkusz Ore Region where a regional depression cone with an area of 400 km$^2$ can be found and the piezometric pressure within the workings of the Zn-Pb ore mines is lowered to 180 m a.s.l.

The typical groundwater of the Olkusz-Zawiercie MGB is fresh, slightly mineralised, and of the $HCO_3$–Ca-Mg, $HCO_3$–$SO_4$ -Ca-Mg types. The water is slightly alkaline (pH 7.0–8.1)

and from soft to very hard (93–1690 mgCaCO$_3$/dm$^3$). The hydrochemical background range (excluding deposit areas) is as follows: HCO$_3$ 160–360; SO$_4$ 26–132; Cl 1.4–13.1; Ca 44–120; Mg 12–50; K 0.5–3.5; Na 1.2–4.9; TDS 190–720 (mg/dm$^3$).

The renewable groundwater resources of the Olkusz-Zawiercie reservoir amounted to 461,260 m$^3$/d (168.36 million m$^3$) and the resource module to 5.2 dm$^3$/s/km$^2$. Groundwater intake from the reservoir in 1994 amounted to 47.4 million m$^3$ from deep water intakes and springs and 21.6 million m$^3$ from mine drainage.

In the area where the unconfined fissured-karst Upper Jurassic reservoir occurs (upper course of the Biała Przemsza valley), vulnerability is high. In the occurrence area of the Olkusz-Zawiercie MGB (middle course of the river), vulnerability is low due to the occurrence of Rhaetic-Kuuper clay formations in the overburden except for the area of the hydrogeological window in the area of the Przemsza fossil valley near Olkusz. The mining area of Zn-Pb ore deposits located in the south of Olkusz is particularly vulnerable to pollution because the rock mass is actively drained and mining exploitation results in fast water circulation [9].

## 3. Results

### 3.1. Development of Zn and Pb Ore Mining in the Olkusz Region and Changes in Groundwater Circulation Conditions Caused by Mining Activity

On the basis of archaeological research in the Biała Przemsza catchment area and the analysis of historical sources (e.g., Georgius Agricola), the beginning of mining and processing of Pb and Ag ores in the Olkusz region was dated to the turn of the 11th and 12th centuries [57,58], and mining privileges were extended by King Władysław Jagiełło in the first half of the 15th century. In the 16th century, five adits were drilled in the vicinity of Olkusz and Bolesław with a total working length of approx. 30 km [8]. The area of the Olkusz and Pomorzany deposits was drained by the following adits: Starczynowska (Królewska; length 3.2 km; outlet to the Sztoła River), Ponikowska (length 5.5 km; outlet to the Biała River), and Pilecka (length 6.0 km, outlet to the Biała River). Drainage of the Bolesław region was provided by the following adits: Czajowska (after reconstruction in the 19th century, Bolesławska; length 3.2 km) and Ostowiecka (Centauryjska; length 4.6 km) connected to the Biała River. The area of Bukowno was drained by the Czartoryska adit (length > 2 km) built at the turn of the 19th and 20th centuries and connected to the Sztoła River. It drained the deposits of the Ulisses and Jerzy mines. In 1674 only two adits were in operation: Ponikowska and Pilecka, mainly as a result of the exploitation of most of the deposits drained due to the presence of the adits [58]. The flooding of the Baba River (a tributary of the Sztoła River) silting up the adits contributed to the decline in mining activity. In 1712, the Ponikowska adit collapsed. Mining activity was limited to the utilisation of old heaps. Exploitation of polymetallic deposits was resumed at the turn of the 18th and 19th centuries after mastering the technology of obtaining and using zinc. In 1814, the Józef mine in Olkusz was established (operating until the beginning of the 20th century); in 1820, the Ulisses mine; in 1821, the Bolesław mine; and in 1823, the Jerzy mine (which merged with the Ulisses mine at the end of the 19th century). The Ostowiecka, Ponikowska, and Starczynowska adits were rebuilt. Washing plants used in the treatment process were built on rivers and water reservoirs. At the end of the 19th century, the installation of steam pumps enabled lowering the water table to an elevation of 285 m above sea level [58,59].

As of 1989, the characteristics of the Zn-Pb ore mines Bolesław, Olkusz, and Pomorzany in the Olkusz region were as follows: mining areas—9.77, 14.01, and 27.25, respectively (in total: 51.03 (km$^2$)); maximum lowering of the water table—150 m; total area of the regional depression cone—400 km$^2$; average annual inflows to mines—20.0, 50.8, and 172.4, respectively (in total: 243.2 (m$^3$/min)) [9].

The volume of mine water drainage from the beginning of the period of adit operation (end of the 16th century) was estimated at 100 m$^3$/min, and due to the limited maintenance of the workings, by the 19th century the inflow decreased to 90–50 m$^3$/min. Around 1880,

the water drained by the Czajowska (Bolesławska) and Czartoryska adits was discharged into the Biała and Sztoła rivers at the rate of 22 m$^3$/min, and in the 1920s—30–33 m$^3$/min. The Franco-Polish Mining Society suspended further mining activity due to the economic crisis in 1931. Under the conditions of self-drainage, the outflow to the watercourses from the mines decreased to 24 m$^3$/min [4]. In 1940, the German company Ostmetall resumed production in ZGH Bolesław (created by the merger of the mines Bolesław and Ulisses). The Mieczysław shaft was sunk and the Południowa (South) Adit driven to be connected with Czartoryska, and mine water drainage increased to 30–33 m$^3$/min [58]. Such levelled inflow lasted for about 30 years. From 1973, in reaction to the commencement of the construction of the Pomorzany Mine, the inflow rapidly decreased to approx. 10 m$^3$/min in 1976 and then increased to 20 m$^3$/min beginning in 1980. In 1996, the drainage of the mining plant was abandoned. With the construction of another plant and accessing the Olkusz deposit in 1957, there was a rapid increase in inflows to 70 m$^3$/min in 1963 and up to 80 m$^3$/min in 1967–1968. In the following years, a downward trend was observed (in particular, again after the start of draining the Pomorzany deposit in 1973) to some 40 m$^3$/min in 2021 with periodic increases in inflows to 60 m$^3$/min (Figure 2).

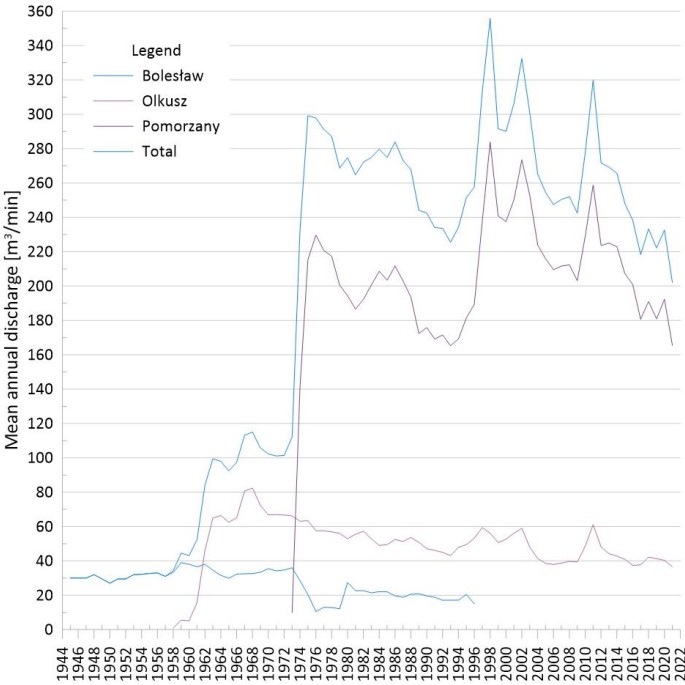

**Figure 2.** Average annual water inflows to drainage systems of Zn-Pb ore mines in the Olkusz region in 1944–2022 after Wilk, Bocheńska eds., 2003 [60], updated with data from ZGH Bolesław, 2023 [51].

Access to the Pomorzany deposit was associated with a rapid increase in inflows in the years 1973–1974 to almost 230 m$^3$/min related to the crossing of the Pomorzany water-bearing fault by mining workings. After accessing the northern and eastern parts of the deposit as well as extreme rainfalls in 1997, the inflows to the mine increased in 1994–1998 to approx. 284 m$^3$/min. The level of annual inflow then gradually decreased to a volume of 165 m$^3$/min with local maximum values (274 m$^3$/min in 2002 and 258 m$^3$/min in 2012) associated with intense precipitation (Figure 2). At the turn of 2021–2022, the drainage system was decommissioned.

### 3.2. Changes in Spatial Development

The analysed part of the Biała Przemsza river catchment was most importantly an area of the centuries-old development of the mining industry. Along with the identification of the next deposits, industry and consequently settlements developed. Mining works involved, among other things, changes in drainage intensity, which in turn affected the

shallow groundwater remaining in hydraulic connectivity. Progressive drainage induced habitat changes. Table 1 and Figure 3A–C show the spatial development in the Biała Przemsza valley in the years 1941, 1961, and 2021.

**Table 1.** Changes in spatial development in the area of Biała Przemsza valley in the period of 1941–2021.

| Landscape Elements | Area in the Following Years (km² (%)) | | |
|---|---|---|---|
| | 1941 | 1961 | 2021 |
| Water reservoirs | 0.95 (0.22) | 0.31 (0.07) | 0.42 (0.09) |
| Built-up areas | 9.51 (2.18) | 14.05 (3.22) | 25.92 (5.95) |
| Industrial areas | 2.08 (0.48) | 2.78 (0.65) | 7.92 (1.82) |
| Wastelands | 4.58 (1.05) | 4.15 (0.95) | 3.03 (0.70) |
| Arable land | 184.10 (42.23) | 187.33 (42.97) | 154.70 (35.48) |
| Wet meadows and peat bogs | 3.34 (0.76) | 6.60 (1.51) | 2.31 (0.53) |
| Meadows and pastures | 43.50 (9.98) | 34.12 (7.83) | 39.64 (9.09) |
| Forests and groups of trees | 162.80 (37.34) | 173.10 (39.70) | 191.87 (44.01) |
| Sands grasses | 25.10 (5.76) | 13.53 (3.10) | 10.16 (2.33) |
| Total area | 435.97 | 435.97 | 435.97 |
| Sum of watercourse lengths (km) | 278.10 | 328.10 | 299.20 |

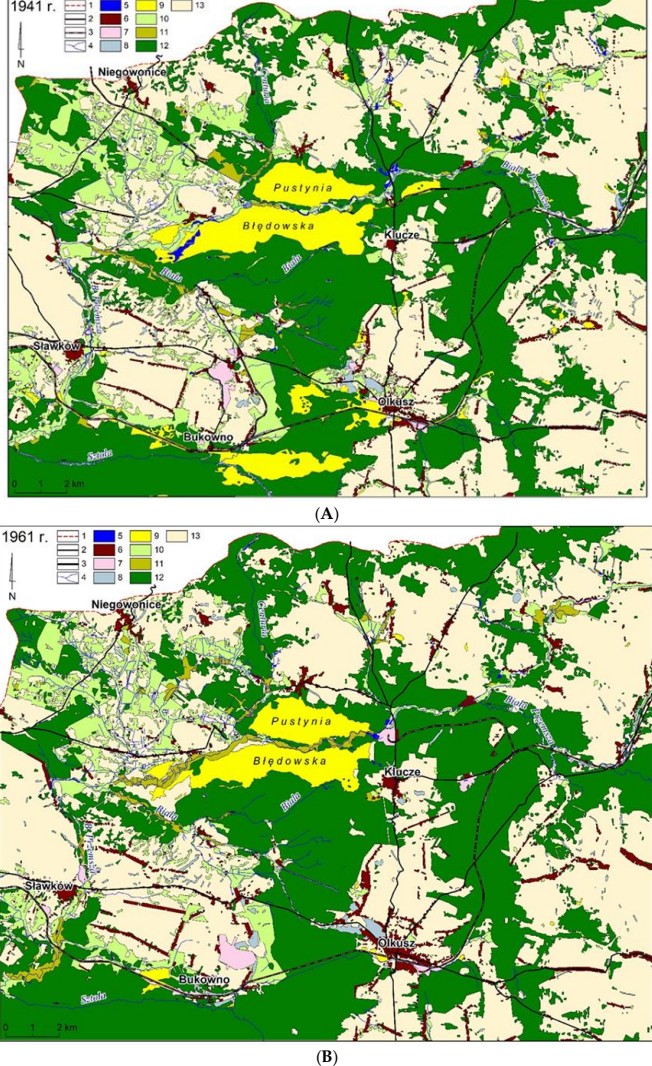

(A)

(B)

**Figure 3.** *Cont.*

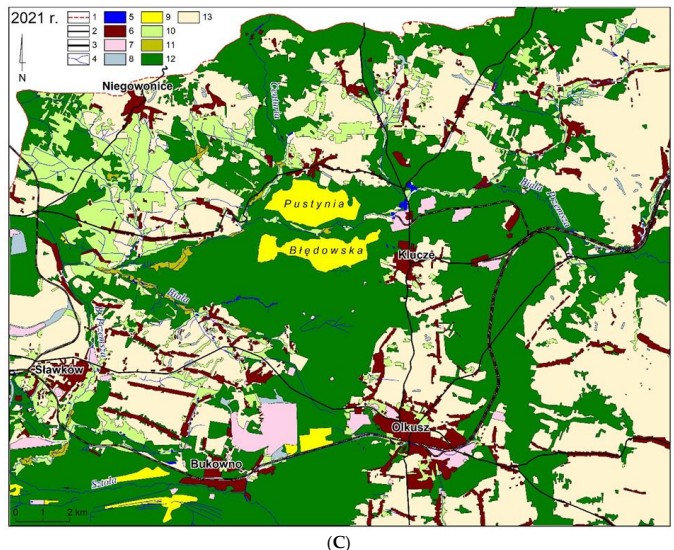

(**C**)

**Figure 3.** (**A**) Changes in land use in the area of Biała Przemsza valley in 1941; (**B**) changes in land use in the area of Biała Przemsza valley in 1961; (**C**) changes in land use in the area of Biała Przemsza valley in 2021. 1—Border of the research area; 2—important roads; 3—important railways; 4—permanent watercourses; 5—water reservoirs and greater river broads; 6—built-up areas; 7—industrial areas; 8—wastelands (industrial and post-industrial, post-agricultural, larger embankments, slopes, gorges, and others); 9—sands; beaches; 10—meadows and pastures; 11—wet meadows and peat bogs; 12—forests and groups of trees; 13—arable land.

As a result of the regional drainage of the Triassic carbonate aquifer after the construction of the Olkusz and Pomorzany mines in the 1960s and 1970s, water conditions changed, resulting in a reduction in the length of the hydrographic network. In the central part of the research area there is the Błędowska Desert, which is accompanied by forest areas to the east and south.

A dense urban and industrial development zone can be found mainly in the southern part of this area. The river network is associated with the Biała Przemsza river flowing in parallel and its tributaries, which are accompanied especially in the western part by meadow and pasture areas as well as wetlands.

According to the conducted analysis, the following trends could be observed: rapid urbanisation and industrialisation in the 21st century as well as a marked reduction in the surface area of arable land and sands. The dynamics of the growth of the area occupied by forests were also observed. The hydrographic network evolved. In the 1960s, its development was related to natural processes (the increase in meadows and pastures accompanied by a decrease in arable land through the formation of bogs and marshes, the emergence of oxbow lakes within the valley, and drainage work (Figure 3B)).

During the periods analysed, there were significant changes in the land use pattern (Table 1) concerning arable land; in 1941, its area was 42.23%, while in 2021, it decreased by 6.75%. A slight increase (1.51%) in wetlands was recorded in 1961 compared to 1941 (0.76%), and a decrease to 0.53% was recorded in 2021. There was a steady increase in forest cover over the study period that ranged from 37.34% (1941) to 39.7% (1961) and then increased to 44.01% in 2021. This increase was partly related to abandoning arable halves, which spontaneously overgrew to form forest communities (Table 1).

### 3.3. Changes in Water Chemistry

Water from the drainage of zinc and lead ore mines flowed through the Dąbrówka Channel to the Biała and through the Południowy Channel to Sztoła, tributaries of the Biała Przemsza. Sztoła flows into the Przemsza west of Bukowno and the Biała in Okradzionów. At the beginning of 2022, drainage was ceased due to the decommissioning of the last mining plant. Mine water discharges were an important element of the Biała Przemsza

recharge after losses related to infiltration along numerous sections within the depression cone. The groundwater came from the drainage of the Lower and Middle Triassic carbonate formations. In the hydraulic connection zone, sections of the Biała Przemsza course and its tributaries changed their character to recharge water as a result of drainage, mixing with groundwater in the workings of the drainage system. The scope of hydrochemical tests performed by the mining services was limited to index analyses during mining activity. The results of physicochemical tests of mine water from 1950 to 2021 are presented in Table 2.

**Table 2.** Variability in selected physical and chemical parameters in groundwater from mines in the Olkusz region (minimum and maximum values; average value in brackets).

| Research Period | 1950–1970 [1] | 1950–1970 [1] | 1950–1970 [1] | 1975 [3] | 1973–1975 [2] | 1996–2020 [5] | 2007–2012 [4] |
|---|---|---|---|---|---|---|---|
| Sampling points | Drilling holes Q | Drilling holes J | Drilling holes T | Collective waters | Dąbrówka Channel/Południowy Channel | Dąbrówka Channel/Południowy Channel | Dąbrówka Channel/Południowy Channel |
| pH (–) | 6.0–8.5 (7.4) | 7.2–8.8 (7.8) | 6.0–8.5 (7.5) | No data | No data | (7.84–8.30)/(7.70–8.30) * | No data |
| $SO_4$ (mg/L) | 13.9–218.9 (65.0) | 3.3–68.7 (26.6) | 16.5–206.0 (40.0) | 21.4–334.9 (50.1) | (74.4–118.3)/(41.3–42.9) * | (177.0–345.0)/(88.0–184.0) * | (290.0–324.0)/(114.0–131.0) * |
| Cl (mg/L) | 4.3–189.0 (22.5) | 5.3–41.5 (11.1) | 1.8–344.4 (14.1) | 3.3–36.2 (11.0) | No data | (10.59–38.96)/(6.98–31.90) * | (22.15–23.51)/(16.81–18.43) * |
| Zn (mg/L) | No data | No data | No data | 0.1–1.0 | –(1.7–2.3)/–(0.8–1.5) * | (0.903–1.690)/(0.587–1306) * | (0.975–1.342)/(0.688–1.068) * |
| Pb (mg/L) | No data | No data | No data | 0.05–0.10 | –(0.8–2.4)/–(0.1–0.8) * | (0.027–0.350)/(0.037–0.293) * | (0.243–0.295)/(0.108–0.201) * |

Sources used: [1] Adamczyk [61]; [2] Różkowski A., Wilk (ed.) [4]; [3] Wilk, Bocheńska (ed.) [60]; [4] Liszka, Świć [58]; [5] ZGH Bolesław [51]. * Average annual concentrations.

Discharged water was for many years selectively pumped and directed to two main outlets in the form of channels: Dąbrówka (for more highly mineralised water) and Południowy (for water of better quality). Details connected with the changing chemistry over time as well as mine water management are presented in the following discussion.

## 4. Discussion

### 4.1. Transformation of the Hydrological Conditions

As a result of mining activity, the water level was lower maximally by 150 m, forming a depression cone reaching up to 400 km$^2$. As a consequence, the Biała Przemsza river changed its character over considerable sections from draining to infiltrating, the quantitative depletion of groundwater resources within the range of the mine drainage cone appeared, the springs of the Biała and Baba rivers disappeared, and the flow of the Biała Przemsza decreased. The most spectacular water escapes from Biała Przemsza were observed after catastrophic rainfall in July 1997 with a downward trend in the Golczowice—Błędowska Desert section (m$^3$/min): 3.622 (5 August 1997), 2.458 (9 September 1997), 1.393 (24 October 1997), 1.279 (4 April 1998), and 0.363 (July 1999) [11].

The water of the Biała River (3.1 km in length) mainly included the groundwater of the Olkusz–Pomorzany Zn-Pb ore mine discharged through the Dąbrówka Channel. In the Biała Przemsza flow below the outlet of the Dąbrówka channel, mine water accounted for 70%. The Baba River (9.9 km in length) flows into the Sztoła in Bukowno. In the section between Olkusz and Bukowno, the Baba River was recharged with water from the drainage of the mining plant through the Południowy (Southern) Channel. In the Sztoła flow below the outlet of the Baba River, mine water accounted for practically 100%.

In 2013–2021, the average annual flows (SQ) in the Niwka water gauge station closing a catchment area of 865.19 km$^2$ were significantly lower and ranged from 5.22 to 7.20 m$^3$/s. At the turn of December 2021 and January 2022, the pumping stations were turned off in the last active Zn-Pb ore mine—Pomorzany in the Olkusz region—and the pumping of water from the workings to the tributaries of the Biała Przemsza River was stopped, which radically reduced the flow of water in the Biała Przemsza. This was documented by the comparison of the average low flow in the river (NQ) for the Niwka water gauge station from 2013–2021 (3.80 m$^3$/s) and from May 2022–August 2022. (average: 0.84 m$^3$/s), after the discharge of mine water was stopped. The projected changes in flows in surface watercourses after the decommissioning of the Olkusz Zn-Pb ore mines are presented in Table 3.

**Table 3.** Projected changes in flows in surface watercourses and surface water quality after the liquidation of the Zn-Pb ore mines.

| Changes in Water Flows in Watercourses in the Initial Period of Reconstruction of the Groundwater Table | |
| --- | --- |
| 1 | Lack of water recharge to Biała Przemsza with mine water flowing through the Dąbrówka Channel through the Biała River; lack of water recharge to Sztoła through the Południowy Channel. |
| 2 | The flow in Biała Przemsza from the connection with Biała towards Sławków will be reduced by an estimated 70–80%; the recharge of mine water to Sztoła will cease and the river will dry up. |
| **Changes in Water Flows in Watercourses in the Further Period of Reconstruction of the Groundwater Table** | |
| 3 | The reconstruction of the groundwater table will take from several to several dozen years and will restore the natural flow of water in Biała Przemsza. |
| 4 | The initial springs of the Biała Przemsza River will appear, and the upper section of the river as well as the dried sections of the Biała Przemsza tributaries will be restored. |
| 5 | Increases in flows to the value of original flows: Biała Przemsza—Golczowice: 0.7–3.0 $m^3/s$; Biała Przemsza—Sławków: 3–8 $m^3/s$. |

### 4.2. Changes in Vegetation Cover as a Result of Impact of Zn and Pb ore Mining and Land Use

Ecohydrological changes and their impact on land use structure and ecosystem functioning under the influence of the Zn-Pb mining should be considered mainly regarding changes in water relations. This refers to the intensive exploitation of Zn-Pb ores by the Pomorzany mine, which began in the 1970s (Figure 1) and caused far-reaching changes in water relations. The drainage works carried out here generally lowered the groundwater level by about 30 m with more tris deposition [62]. As a result of this process, the Quaternary-Jurassic aquifer disappeared completely in zones of low thickness, and the groundwater table lowered relatively quickly. The upper section of the Biała stream (Figure 1) and other smaller streams utterly disappeared, and water escapes were observed on the Biała Przemsza [4]. Drastic changes in water relations have caused changes in riparian ecosystems developing in marshy and wet areas. Riparian communities from the Alno-Padion association have entirely disappeared, and their areas are now overgrown by pine forests in the eastern fragment of the "Pustynia Błędowska" [63] where the hygrophilous species Chochlearia polonica (endemic to Poland) has disappeared [62]. The artificially lowered groundwater level influenced the shrinkage of meadows of the Molinion union in the western and northern parts of the study area (Figure 3C). A similar phenomenon was found in other areas influenced by mines leading to lower groundwater levels [64–68].

Vegetation ecosystems have undergone significant changes during the last 80 years due to anthropogenic pressure. This effect is observed as mentioned mostly in the case of wet meadows—moist and marshy meadows that result from the lowering of groundwater levels within the depression cone created by mining activities in the area. The meadows belong to the associations of *Calthion* and *Molinion caerulea* represented by the following communities: *Galioveri-Molinietum* and *Junco-Molinietum*. Their respective areas increased in 1961 (1.51%)

and decreased in 2021 (0.53%) (Table 1, Figure 3C). Non-forest communities of the classes *Phragmittea*, *Scheuchzerio-Caricetea fuscae*, *Artemisietea*, and *Molinio-Arrhenatheretea* also occur.

These areas are now mainly occupied by willow meadows of the *Salicion cinereae* and *Sambuco-Salicion capreae* associations. Even before 1941, wet meadows in the valley of the Biała and Biała Przemsza rivers (the largest areas were occupied in the period of 1941–1961—Figure 3A,B) were regularly mowed and grazed, which resulted in the inhibition of forest succession [63,69]. On the other hand, changes in water relations in the Biała Przemsza catchment caused by the activities of zinc and lead mines led to a decrease in groundwater levels and caused a transition from wet meadows to fresh or dry xerothermic grasslands.

Extensively used lowland mesophilous meadows occur in the northwestern part of the studied catchment area (Figure 3C) and have developed on potential oak–hornbeam (*Carpinion*) and light-woodland habitats as a result of logging for crop production and animal farming. The slight change in their area (9.09%; Table 1) is related to forest succession, and further transformation of this phytocenosis will depend on the active protection of meadows because a part of the described habitats is protected under the NATURA 2000 scheme.

The most interesting, well-documented area of transformation of the biotic environment in the described area is the so-called "Pustynia Błędowska" (Błędowska Desert) (Figure 3A–C). This is an anthropogenic desert created due to deforestation for fuel material used in processing and mining in the Middle Ages. On topographic maps from 1804 to 1933, the Błędowska Desert is depicted as an area devoid of vegetation [63,70]. The area was subject to constant successional changes related to aeolian phenomena and regular military exercises. In the first and second cases, vegetation destruction occurred. Once the disturbance factor ceased, succession continued. Spontaneous succession in the Błędowska Desert occurs according to the classical succession model, the model of changing resource coefficients of Tilman [71], and the facilitation model [2,24].

During these periods and currently, sand grasslands with the main contribution of psammophilous species such as *Corynephorus canescens*, *Koeleria glauca*, *Festuca psammophila*, and *Elymus arenaria* develop on open sandy areas with loose sand. These are protected grasslands within the scope of NATURA 2000 sites. They are widespread in the form of an insular range in the subcontinental and continental parts of central Europe but are less frequent in other regions of the continent and are therefore protected [72]. After building the Katowice Steelworks in 1972, a so-called forest protection zone was created around it. As part of the planned afforestation, the entire western part of the Błędowska Desert was planted with tree species. By the 1990s, native trees were introduced: *Pinus sylvestris*, *Betula pendula*, *Alnus incana*, and *A. glutinosa*; as well as alien trees: *P. nigra*, *P. strobus*, *Robinia pseudacacia*, and *Quercus rubra* [63,73].

A steady increase in forest cover was observed over the study periods. In 2021, it occupied 44.01% of the catchment area (Table 1, Figure 3C). The increase in forest cover is due to the overgrowth of forests on mainly uncultivated land and the overgrowth of meadows. The catchment area of the Biała Przemsza River is located on a mosaic of diverse geological formations (sands, clays, loess, and limestone), which affects the type of forest ecosystems found in the area. The dominant communities here *are Leucobryo-Pinetum*, *Querco roboris-Pinetum*, and patches of *Cladonio-Pinetum*. Beech forest (*Dentario enneaphylli-Fagetum* and *Luzulo pilosae-Fagetum*) and *Tilio-Carpinetum* forests develop on limestone and loess substrates. Along the watercourses in the catchment, there are in turn deciduous forests (alder forests; less frequently riparian forests) *Alnion glutinosae*, *Alno-Ulmion*, and willow thickets belonging to the class *Salicetea purpureae* [36,74].

*4.3. Changes in Water Chemistry as a Result of Development of Exploitation of Zn and Pb Ore Deposits*

In the years 1950–1970; i.e., in the period preceding the intensive development of mining operations, the water of the Triassic aquifer was characterised by a wide range

of variability in the concentrations of SO$_4$ (16.5–206 mg/L; average 40 mg/L) and Cl (1.8–344 mg/L; average 14.1 mg/L) (Table 2), which indicates very diverse environmental conditions [61].

In collective water tested before 1975, the average concentrations of SO$_4$ and Cl were 50.1 and 11.0 mg/L, respectively, and the contents of Zn and Pb were 0.1—1.0 and 0.5—0.10 mg/L, respectively. Systematic testing of the water flowing into the workings of the Olkusz and Pomorzany mines began in 1976. Since then, along with the development of mining in the region, the chemical composition of drained water has been altered [60].

Along with the expansion and increasing the exploitation depth of Zn-Pb ores, the depression cone expanded, exposing further parts of the orogen to the oxidation effect, while covering and inducing the surface of pollution sources, especially post-flotation sedimentation ponds and the spillway of liquid waste of the paper mill in Klucze on the eastern edge of the Błędowska Desert (the Przemsza proglacial valley filled with clastic deposits).

In the years 1973—1975, as part of selective water management, high-quality water already was directed to the Południowy Channel and low-quality water to the Dąbrówka Channel. The water of the Południowy Channel was characterised by the following average concentrations of components—SO$_4$: 41.3–42.9; Zn: 0.8—1.5; Pb: 0.1—0.8 (mg/L). In the water of the Dąbrówka Channel, the average concentrations of components were usually twice as high—SO$_4$: 74.4–118.3; Zn: 1.7–2.3; Pb: 0.8–2.4 (mg/L) [4] (Table 2).

The year 1973 saw the beginning of the operation of the Pomorzany Mine, which subsequently drained northeastern parts of the deposit, while the mining deepened to approximately 140 m below ground level. In older mines, the maximum mining depth was: Bolesław—60 m b.g.l.; Olkusz—90 m b.g.l. Consequently, the depressed zone expanded significantly, the inflows to the mines increased, and significant parts of the orogen were drained and aerated. As a result, the average annual concentrations of SO$_4$ in the years 1996—2021 increased and amounted to the following: in the Południowy Channel—88–184 mg/L (median 117 mg/L); and in the Dąbrówka Channel—177–345 mg/L (median 268.5 mg/L) (Table 2, Figure 4).

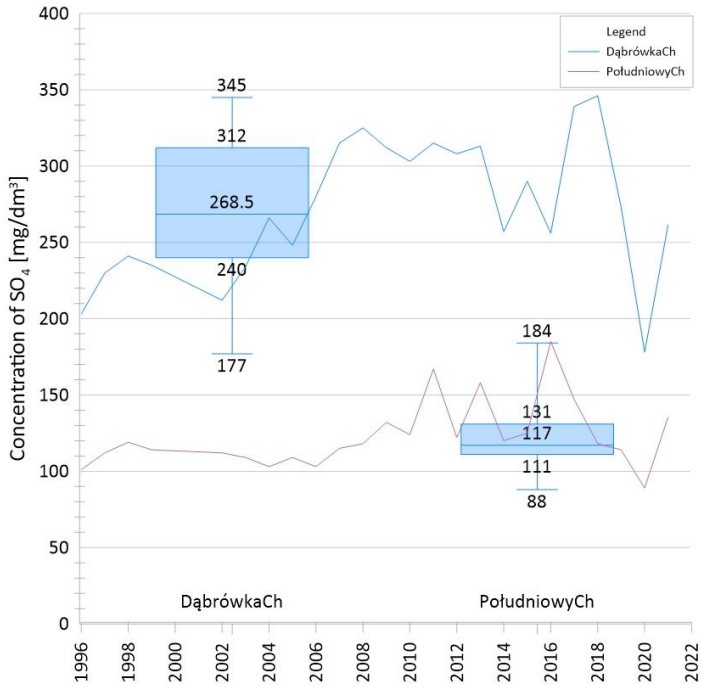

**Figure 4.** Variation in the average annual concentrations of sulphates in mine water discharged to Biała Przemsza in 1996–2022 (data: ZGH Bolesław [51]).

Cl concentrations showed greater fluctuations in average values for the Południowy and Dąbrówka Channels within the ranges of 6.98–31.9 (median 18.98) and 10.59–38.96 (median 23.78) (mg/L), respectively, with abrupt incidental increase in concentrations to above 30 mg/L associated with the 1997 flood. A stable upward trend in concentrations was observed (Table 2, Figure 5).

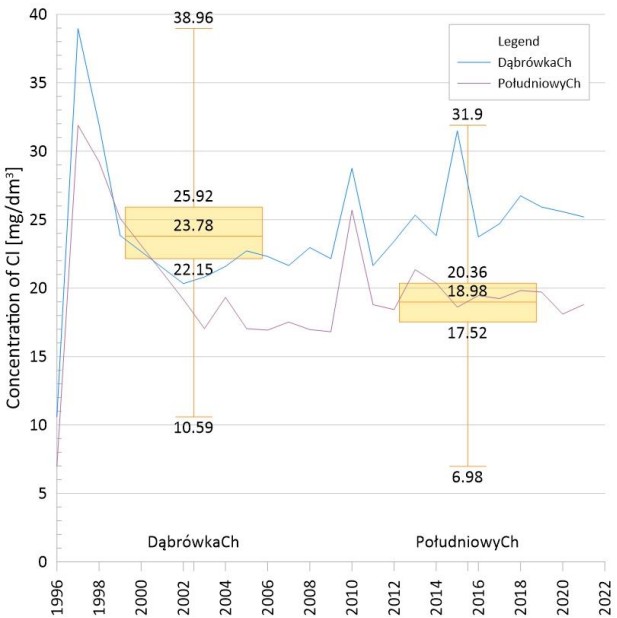

**Figure 5.** Variability of the average annual concentrations of chlorides in mine water discharged to Biała Przemsza in 1996–2022 (data: ZGH Bolesław, [51]).

Concentrations of Zn and Pb in water compared to those in the 1970s have decreased, particularly for Pb (0.037–0.293 mg/L in the Południowy Channel and 0.02–0.350 mg/L in the Dąbrówka Channel, with different average values in the subpopulations of 0.1125 and 0.226 mg/L, respectively) (Figure 6).

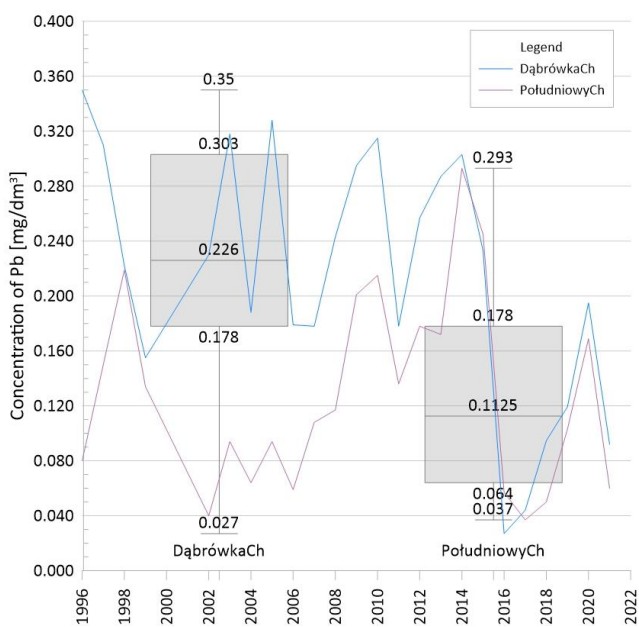

**Figure 6.** Variation in the average annual Pb concentrations in mine water discharged into Biała Przemsza in 1996–2022 (data: ZGH Bolesław, [51]).

The average concentrations of Zn in the water of the Południowy and Dąbrówka Channels were 0.587–1.306 and 0.903–1.690 mg/L, respectively, with median values of 0.949 and 1.250 mg/L, respectively (Figure 7).

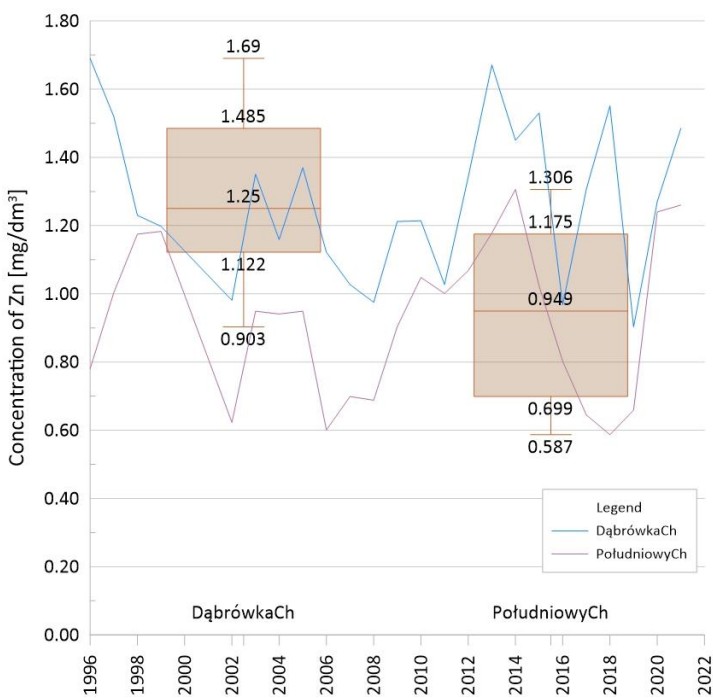

**Figure 7.** Variation in the average annual concentrations of Zn in mine water discharged to Biała Przemsza in 1996–2022 (data: ZGH Bolesław, [51]).

The observed decreases in the concentrations of Zn and Pb in mine water in the 1990s are related to the inflow of clean water from the Pomorzany region, especially its northern section, where there was no mining activity in the past. The inflowing water did not wash the "dried parts" of the orogen exposed to many years of oxidation processes, especially in the area of the Bolesław Mine. With the predominance of the inflow from previously unexploited regions, the concentrations of metals in the collecting water were diluted, while the average concentrations decreased but not the load.

The hyporheic zone of the Przemsza River is a place of transformation and transport of pollutants from Zn and Pb ore mines [75]. High contents of Mn, Pb, Cd, and Zn (locally Tl) were confirmed in the water of the Biała Przemsza River. In bottom sediments in Sławków, the content of Pb reached 6000 mg/kg. The studies showed that As, Sb, and Cr were bound mainly to oxides, organic matter, and sulphides in bottom sediments. BCR extraction of bottom sediments showed that Cd and Zn were associated with cations/anions and carbonates loosely adsorbed on bottom sediments in spring and summer [6,76]. The chemistry of surface water and sediments as conditioned by the influence of mining of Zn and Pb ores is also presented in other publications [21,77,78].

After the cessation of pumping in the Zn-Pb ore mines in Olkusz, it is predicted that below the sites of former mine water discharges, the water quality will improve and the total mineralisation of water and the concentration of $SO_4$ (up to several dozen mg/L) as well as Fe, Zn, and Pb will decrease. In the longer term, as the depression cone fills, polluted groundwater will begin to recharge Biała Przemsza (from Okradzionów to Sławków), and the water quality in the river will deteriorate ($SO_4$ concentrations may reach up to 300 mg/L).

*4.4. Use of Mine Water as an Element Influencing the Change in Hydrological Conditions and Water Management in the Catchment Area of Biała Przemsza*

Since the Middle Ages, mine water has been used for processing and technological purposes (for washing plants and water wheel powering; for example, bellows in mills). The consumption of water in washing plants increased during the period of utilisation of heaps and the start of zinc mining at the end of the 19th century.

Due to the growing demand for good quality water in the urbanizing industrial area of Silesia to the west, the construction of the Maczki intake in Biała Przemsza began in 1929 east of Sosnowiec. Two years later, the production reached 40,000 $m^3$/d, and in 1937 an intake was built on the Sztoła River in Ryszka [79]. The intakes were recharged by mine water from the Południowy Channel. In 1955, another intake was built on the Centralny Channel, which drains water by gravity from the neighbouring Szczakowa Sand Mine. The intakes on the channel and the Biała Przemsza supply drinking water to the surrounding towns.

In the 1970s, out of the 300 $m^3$/min pumped out of the Olkusz mines, 12 $m^3$/min was used for supplying Olkusz waterworks, 25–30 $m^3$/min was used for the mines' own needs (including the processing plant, steelworks, and sulphuric acid production plant), and over 250 $m^3$/min was discharged through channels to the catchment area of the Biała Przemsza (being partly captured again at the intake in Ryszka) [4].

The water drained by the Bolesław Mine after the expansion of the Bolesław Mining and Metallurgical Combine was taken over for the process needs of the plant. Water from the Olkusz Mine pumped out through the Bronisław and Stefan shafts was used following selective management to supply Olkusz with drinking water and partly for the purposes of flotation, and the remaining part was directed through the Południowy Channel to the Baba River flowing into the Sztoła River and through the Roznos and Dąbrówka Channels to the Biała River [60].

Water from the Pomorzany Mine pumped through the Chrobry, Mieszko, and Dąbrówka shafts was discharged in the 1970s into the Dąbrówka and Południowy Channels. Water flowing into the Chrobry Shaft from the area isolated from the surface by Keuper formations had a mineralisation of 0.4–0.6 $g/dm^3$. Water from the remaining shafts was of low quality due to the impact of infiltration of water from post-flotation ponds (Mieszko shaft; mineralisation up to 1.2 $g/dm^3$) and additionally processed wastewater from the discharge of liquid paper production waste from the plant in Klucze (Dąbrówka shaft) and was directed exclusively into the Dąbrówka Channel. Thanks to the selective water management and directing clean water to the Południowy Channel, part of the water was collected until 2017 for water supply purposes in Olkusz.

In the 1990s, the use of mine water increased. With an average annual inflow of 256.4 $m^3$/min (1996), 20 $m^3$/min were taken for drinking purposes and 27.3 $m^3$/min for technological purposes, while some 60 $m^3$/min were captured at the surface water intake in Ryszka on the Sztoła River [60].

## 5. Conclusions

During the period of mining activity between 1941 and 2021, the natural environment of the Biała Przemsza valley underwent various changes.

1.  Mining of Pb and Ag (later also Zn) ores in the Olkusz region began in the 11th century. After the construction of the Olkusz and Pomorzany Zn-Pb ore mines (from 1957 to 1975), regional drainage of the Triassic aquifer complex intensified (as of 1989, the maximum lowering of the water table was 150 m and the total area of the regional depression cone was 400 $km^2$).
2.  The following trends in spatial development in the Biała Przemsza valley could be observed: rapid urbanisation and industrialisation in the 21st century, marked reduction in the surface area of arable land and sands, and dynamic growth of the surface area occupied by forests. The hydrographic network evolved.

3. Between 1941 and 2021, vegetation ecosystems changed as a direct result of anthropogenic pressure. Lowering of groundwater levels due to mining activity resulted in a transformation from wet meadows to fresh or dry xerothermic grasslands. Forest cover increased (in 2021, it occupied 44% of the catchment area). The Błędowska Desert was devoid of vegetation until the 1930s; currently, its western part is forested, and sand grasslands appear on the loose sands.

4. Changes in the hydrodynamic conditions have resulted in the diversity of the hydrogeochemical regime of the Triassic aquifer, which has been manifested by increased concentrations of, e.g., $SO_4$ and Cl. The discharge of lignosulphonate compounds from the paper mill caused periodic degradation of part of the water flowing into the Pomorzany mine.

5. As a result of mining activities, the Biała Przemsza River changed its character to an infiltrating one, a quantitative depletion of groundwater resources in the range of the depression cone of the mine drainage occurred, and there was a reduction in the flow of the Biała Przemsza River. At the same time, mine water was discharged into the Biała and Sztoła Rivers—tributaries of the Biała Przemsza. Further radical reduction in the flow of the Biała Przemsza and its tributaries occurred after the decommissioning of Zn-Pb ore mines and stopping the mine water discharge.

6. Forecasted changes in surface water after the liquidation of the Olkusz Zn-Pb ore mines: ending of the Biała Przemsza water recharge with mine water flowing through the Biała and Sztoła rivers will result in restoration of natural water flow in the Biała Przemsza River, while water quality in the river will ultimately deteriorate (concentrations of, e.g., $SO_4$ will increase).

**Author Contributions:** Conceptualization, K.R., J.R. and O.R.; methodology, K.R., J.R. and O.R.; software, K.R., J.R. and O.R.; formal analysis, O.R.; investigation, K.R., J.R. and O.R.; resources, K.R.; data curation, J.R.; writing—original draft preparation, K.R., J.R. and O.R.; writing—review and editing, O.R.; All authors have read and agreed to the published version of the manuscript.

**Funding:** This research received no external funding.

**Institutional Review Board Statement:** Not applicable.

**Informed Consent Statement:** Not applicable.

**Data Availability Statement:** Data will be made available directly by the authors upon request.

**Conflicts of Interest:** The authors declare no conflict of interest.

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
