# Peer review of "The Historical Aspect of the Impact of Zn and Pb Ore Mining and Land Use on Ecohydrological Changes in the Area of the Biała Przemsza Valley (Southern Poland)"

_land, doi:10.3390/land12050997_

Round 1

Reviewer 1 Report

The manuscript is interesting, but some problems are in it. The details are as below:

1. The Introduction part should be enhanced.

2. Lines 95, 131, 145, and 161, the number of the title should be checked.

3. Lines 118-120, the sentence should be reorganized, and some errors appear in it.

4. The abbreviation should provide the full name before it appears. For example, line 119, “S”; line 121, “SW”.

5. I suggest using the land use transfer metric to show the land use conversion, which would be more clear than the there figure.

6. The connection between land use changes, Zn, Pb ore mining, and ecohydrological changes should be enhanced, better explain how they affect the ecohydrological.

7. The conclusions should be shortened, and only show the core conclusions.

Some descriptions should be modified to make the manuscript more readable.

Author Response

Dear Reviewer and Editor.

At the outset, we would like to thank you for your constructive comments on our article, which made it better than the original version. All reviewers' comments have been considered and are highlighted in red in the main text.  As suggested by the Editor-in-Chief, we have changed the structure of the chapters. This concerned the results and discussion chapters. These are now separate subsections.

Below are the responses to the individual comments.

Reviewer 1.

The Introduction part should be enhanced.

Authors: Thank you very much for your pertinent comment; the introduction has been slightly modified and expanded.  Such fragment was inserted: „After the II WW, three mines were commissioned, the last of which completed dewatering in 2021. The development of lead and silver ore, and then zinc mining in the Olkusz region, became the driver of changes in hydrological conditions [4,5,6,7]. As early as in the 16th century, five adits for draining ore deposits were driven in the region [8], with a total length of workings of approx. 30 km, and in the 1980s, around the Zn-Pb mines, the total regional depression cone already boasted an area of 400 km2 [9], or according to others, even 500 km2 [10]). The maximum total inflows to the mines reached 360 m3/min in the 1990s, and mine water was discharged mainly to watercourses in the catchment area of the Biała Przemsza [9,11]. Research on the impact of mining operations on the water environment was also conducted [5, 12-18]. Gradual lowering of the drainage base to 60, 90 and finally 140 b.g.l. Influenced widening area, particularly strongly affecting wetlands within river valleys. Large depressions of the water table can reach neighbouring hydrogeological units, possibly neighbouring water horizons, which function as separate subsystems [19].”

Reviewer 1.

  1. Lines 95, 131, 145, and 161, the number of the title should be checked.

Authors: Thanks for attention. We have checked and corrected it.

Reviewer 1.

  1. Lines 118-120, the sentence should be reorganized, and some errors appear in it.

Authors: We reorganized it, and now it is correct.

Reviewer 1.

  1. The abbreviation should provide the full name before it appears. For example, line 119, “S”; line 121, “SW”.

Authors: We corrected it and in red colour in the main text.

Reviewer 1.

  1. I suggest using the land use transfer metric to show the land use conversion, which would be more clear than the there figure.

Authors: Thank You very much for comments in this issue. We felt that land cover/land use in tabular form is a good form and, for future reference by other authors, is easier. For this, we would like to stay with it.

Reviewer 1.

  1. The connection between land use changes, Zn, Pb ore mining, and ecohydrological changes should be enhanced, better explain how they affect the ecohydrological.

Authors: Thank you very much for your constructive comments. This is exactly what was missing, and the text has been improved in this respect. Such a fragment was inserted into the main text: " Ecohydrological changes and their impact on land use structure and ecosystem functioning under the influence of the Zn-Pb mining should be considered mainly regarding changes in water relations. This refers to the intensive exploitation of Zn-Pb ores by the Pomorzany mine, which started in the 1970s (Figure 1) and caused far-reaching changes in water relations. The drainage works here generally lowered the groundwater level by about 30 m and more tris deposition [62]. As a result of this process, the Quaternary-Jurassic aquifer disappeared completely in zones of low thickness, and the groundwater table lowered relatively quickly. The upper section of the Biała stream (Figure 1) and other smaller streams disappeared utterly, and water escapes were observed on the Biała Przemsza [4]. Drastic changes in water relations have caused changes in riparian ecosystems developing in marshy and wet areas. Riparian communities from the Alno-Padion association have entirely disappeared, and their areas are now overgrown by pine forests in the eastern fragment of the "Pustynia Błędowska" [63], where the hygrophilous species Chochlearia polonica, endemic to Poland, has disappeared [62]. The artificially lowered groundwater level influenced the shrinkage of meadows of the Molinion union in the western and northern parts of the study area (Figure 3C). A similar phenomenon was found in other areas influenced by mines leading to lower groundwater levels [64, 65,66, 67, 68].

Reviewer 1: 7. The conclusions should be shortened, and only show the core conclusions.

Authors; the conclusion was shortened a little. We hope it's Ok now.

Reviewer 2 Report

The paper by Kazimierz Różkowski etc. entitled “The historical aspect of the impact of Zn and Pb ore mining and land use on ecohydrological changes in the area of the Biała Przemsza valley (southern Poland)” is an interesting study on the impact of Zn and Pb ore mining and land use on ecohydrological changes in the area of the Biała Przemsza valley from 1941 to 2021. The authors investigated the land use making use of GIS. They found the trends observed in the spatial development of the Biała Przemsza valley. The study is well conducted, and the methods used are appropriate. The data is presented clearly. These findings will be of interest to researchers and extension workers, etc., in the field. 

I have the following minor concerns.

1)     P.1L.4 Title: “Biala Przemsza valley” or “Biala Przemsza Valley”? Please clarify.

2)     P.1L.13-32 Abstract: Please clarify the method. For example, GIS was used for the map analysis.

3)     P.1L.30 There are some English grammatical error. “were” should be revised to “was.”

4)     P.1L.31 “occured” should be revised to “occurred.”

5)     P.2L.51 “were” should be revised to was.

6)     P.2L.81 “was” should be revised to “were.”

7)     P.2L.85 There are some typographical errors. “orthophotomap” should be revised to “orthophoto map.”

8)     P.2 L.81-93 These two paragraphs have just a few sentences and should be integrated to one paragraph.

9)     P.3 L.96-102 These two paragraphs have just a few sentences and should be integrated to one paragraph.

10) P.3L.101 “macroregion” should be revised to “macro region.”

11) P.3L.111 “level” should be deleted.

12) P.4L.172 and 179 “are” should be revised to “is.”

13) P.4L.178-P.5L.186 These two paragraphs have just a few sentences and should be integrated to one paragraph.

14) P.5L197, L209, L221 Appropriate indent should be aligned.

15) P.7L.215 “1941” should be revised to “1961.”

16) P.8L.241 “1941” should be revised to “2021.”

17) P.10 L351 P.13L.437 P.14L.465 L475 P.15 L.485 Lines in Figures 3 to 7 should be clearly displayed.

18) P.11L.390 “the years” are wordy and should be deleted.

19) P.12L.417 “the period” are wordy and should be deleted.

20) P.12L.427 “were” should be revised to “was.”

21) P.13L.435 “was” should be revised to “has been.”

22) P.13L.455 Table. 3, Fig. 4 should be revised to Table 3, Figure 4.

23) P.16L.535 were should be revised to was.

24) P.17L.577 have should be revised to has.

Author Response

Dear Reviewer and Editor.

At the outset, we would like to thank you for your constructive comments on our article, which made it better than the original version. All reviewers' comments have been taken into account and are highlighted in red in the main text.  As suggested by the Editor-in-Chief, we have changed the structure of the chapters. This concerned the results and discussion chapters. These are now separate subsections.

Below are the responses to the individual comments.

Reviewer 2.

P.1L.4 Title: “Biala Przemsza valley” or “Biala Przemsza Valley”? Please clarify.

Authors; Thank you for your attention, we clarify in the text. Now we have the Biała Przemsza valley everywhere.

Reviewer 2.

2)     P.1L.13-32 Abstract: Please clarify the method. For example, GIS was used for the map analysis.

Authors: Thank You for the suggestion. We added.

Reviewer

 3)     P.1L.30 There are some English grammatical error. “were” should be revised to “was.” OK

4)     P.1L.31 “occured” should be revised to “occurred.”

5)     P.2L.51 “were” should be revised to “was.”

6)     P.2L.81 “was” should be revised to “were.”  

Authors: We corrected it according the reviewer's suggestions.

Reviewer

7)     P.2L.85 There are some typographical errors. “orthophotomap” should be revised to “orthophoto map.”

Auhtors: Revised and corected.

8)     P.2 L.81-93 These two paragraphs have just a few sentences and should be integrated to one paragraph.

9)     P.3 L.96-102 These two paragraphs have just a few sentences and should be integrated into one paragraph.

Authors: We integrated it in red colour in the main text.

10) P.3L.101 “macroregion” should be revised to “macro region.”

Authors; Revised and corrected

Reviewer

11) P.3L.111 “level” should be deleted. OK

Authors: level deleted

Reviewer 2

12) P.4L.172 and 179 “are” should be revised to “is.” OK

Authors: revised and corrected

Reviewer

13) P.4L.178-P.5L.186 These two paragraphs have just a few sentences and should be integrated to one paragraph.

Authors; Thank you very much for your correct comment. These two paragraphs deal with slightly different content of the problem, and we have left them separate. We hope that in this form, it will not be so problematic.

Reviewer

14) P.5L197, L209, L221 Appropriate indent should be aligned. OK

Authors: revised and corrected

Reviewer

15) P.7L.215 “1941” should be revised to “1961.”

16) P.8L.241 “1941” should be revised to “2021.”

Authors; We revised and corrected changes in the red colour

Reviewer

17) P.10 L351 P.13L.437 P.14L.465 L475 P.15 L.485 Lines in Figures 3 to 7 should be clearly displayed.  

Authors: We modified the title of figures 3-7, and now more clearly. Thanks for the comments.

Reviewer

18) P.11L.390 “the years” are wordy and should be deleted.

19) P.12L.417 “the period” are wordy and should be deleted.

20) P.12L.427 “were” should be revised to “was.”

21) P.13L.435 “was” should be revised to “has been.”

22) P.13L.455 “Table. 3, Fig. 4” should be revised to “Table 3, Figure 4.”

23) P.16L.535 “were” should be revised to “was.” 

24) P.17L.577 “have” should be revised to “has.” 

Authors; All the comments listed above have been amended and corrected. Changes have been made to the main text.

Round 2

Reviewer 1 Report

The manuscript is much improved after revision, it could be publicated after minor revise. The details are as below:

1. Line 373, the km2 should be modified.

2. The font of the body needs to be unifrom.

Moderate editing of English language is needed.

Author Response

We are very grateful for the detailed review, it makes the text very clear. 
Reviewer:
1. line 373, km2 should be modified.
Authors; Corrected

Reviewer: 2. The font of the corpus should be standardised.
Authors; We standardised some elements that were different in nature.

Before we submitted the text for review, it was corrected by a native speaker. Today he checked it again and confirmed that it is fine. As we are not native speakers we have to believe that it is fine.